# Saffron Extract Attenuates Anxiogenic Effect and Improves Cognitive Behavior in an Adult Zebrafish Model of Traumatic Brain Injury

**DOI:** 10.3390/ijms231911600

**Published:** 2022-10-01

**Authors:** Victoria Chaoul, Maria Awad, Frederic Harb, Fadia Najjar, Aline Hamade, Rita Nabout, Jihane Soueid

**Affiliations:** 1Department of Life and Earth Sciences, Faculty of Sciences, Lebanese University, Jdeidet P.O. Box 90656, Lebanon; 2Department of Biomedical Sciences, Faculty of Medicine and Medical Sciences, University of Balamand, Kalhat, Al Kurah P.O. Box 100, Lebanon; 3Laboratoire d’Innovation Thérapeutique, Departments of Biology, Chemistry and Biochemistry, Faculty of Sciences II, Lebanese University, Jdeidet P.O. Box 90656, Lebanon; 4Department of Biochemistry and Molecular Genetics, Faculty of Medicine, American University of Beirut, Beirut P.O. Box 11-0236, Lebanon

**Keywords:** traumatic brain injury, saffron, zebrafish, anxiolytic, fear disinhibition, memory performance

## Abstract

Traumatic brain injury (TBI) has the highest mortality rates worldwide, yet effective treatment remains unavailable. TBI causes inflammatory responses, endoplasmic reticulum stress, disruption of the blood–brain barrier and neurodegeneration that lead to loss of cognition, memory and motor skills. Saffron (*Crocus sativus* L.) is known for its anti-inflammatory and neuroprotective effects, which makes it a potential candidate for TBI treatment. Zebrafish (*Danio rerio*) shares a high degree of genetic homology and cell signaling pathways with mammals. Its active neuro-regenerative function makes it an excellent model organism for TBI therapeutic drug identification. The objective of this study was to assess the effect of saffron administration to a TBI zebrafish model by investigating behavioral outcomes such as anxiety, fear and memory skills using a series of behavioral tests. Saffron exhibited anxiolytic effect on anxiety-like behaviors, and showed prevention of fear inhibition observed after TBI. It improved learning and enhanced memory performance. These results suggest that saffron could be a novel therapeutic enhancer for neural repair and regeneration of networks post-TBI.

## 1. Introduction

Traumatic brain injury (TBI) [1], represents a major cause of morbidity and mortality worldwide leading to temporary or permanent impairment [2]. The Centers for Disease Control and Prevention (CDC) reported that TBIs accounted for approximately 2.2 million emergency department visits, 223,000 hospitalizations and 64,000 deaths in the United States in 2020 [2,3]. Defined as a sudden and brutal injury to the brain [1], TBIs are caused by a mechanical stabbing or acceleration/deceleration waves forcing the brain to move rapidly within the skull [4], resulting in hemorrhage, contusion and immediate clinical effects [5]. It has become of great concern due to its complex and multifaceted pathogenesis [6] which plays a major role in the paucity of proven therapies for this “silent epidemic” [4]. Following brain injury, a primary insult takes place, characterized by tissue deformation, blood–brain barrier (BBB) damage and cerebral fluid increase resulting in an increased intracerebral pressure (ICP) [7]. An elevation in glutamate levels arises as a secondary insult [8], leading to an increased level of intracellular calcium. This latter factor notably activates caspases and a cascade of free radicals, which results in excitotoxicity, edema, neuroinflammation and apoptosis [9,10]. These effects are associated with long-lasting cognitive decline including post-traumatic amnesia, loss of consciousness, motor deficiencies and anxiety-like behavior [11,12,13,14,15], manifesting within days, weeks, or even years after initial injury [16]. The relative degree of each aforementioned factor is the primary challenge facing clinicians in TBI treatment.

Prevention of the secondary injury is the utmost target in any treatment of TBI [17]. However, despite extensive efforts to develop neuroprotective therapies for TBI patients, there are no effective neuroprotective agents to improve TBI-related psychosocial sequalae. This is attributed to the unsuccessful translation of experimental studies into the clinical practice [18]. Although many studies have described pharmacological treatment targets to enhance recovery post-TBI, data remain controversial as the administration of a combination of compounds post-injury have aggravated the outcome in various cases [19]. Therefore, investigations of natural compounds showing low toxicity levels as a therapeutic intervention following injury have emerged [20,21,22].

Recent evidence supports a relevant role for *Crocus sativus* L., commonly known as saffron, as a new pharmacological target and an enhancer of cognitive functions [23]. *Crocus sativus* L. is a dietary perennial stemless herb belonging to the family of Iridaceae [24]. It is mainly produced in Iran and cultivated in other countries such as France, Greece, Italy, Spain, India, China, Turkey and Lebanon [25]. The four major bioactive compounds of saffron are crocin, picrocrocin, safranal and crocetin. It also contains water, nitrogenous matter, sugars, fibers, soluble extracts, volatile oil, vitamins (riboflavin and thiamine), and more than 150 volatiles and aroma-yielding compounds [25,26]. Crocin is the main anti-oxidant constituent of saffron as a dye material proven to be responsible for the health-promoting properties of saffron [27]. *Crocus sativus* L. has been studied in many animals’ disease models. Studies have shown that saffron has a broad spectrum of pharmacological properties, including anti-hyperglycemic [28], anti-depressive [29], anti-tumor [27], and anti-oxidant [30] effects that have been identified in rodent disease models. Specifically, administration of crocin in a controlled cortical impact (CCI) model of TBI in mice showed a decrease in brain edema and neurological motor severity score [31]. Apoptotic cell death and neuroinflammation were inhibited in a TBI-induced mouse model treated with crocin, through the Notch signaling pathway [17].

In adult mammals, extracellular and intracellular factors within the central nervous system (CNS) hinder its neuronal regeneration capabilities following an injury [32]. Therefore, it is of utmost importance to search for effective animal models that could delineate the basis of the neurodegeneration and evaluate the effect of novel drugs in the CNS recovery to its prior functional state. With a high genetic homology to the human reference genome, zebrafish (*Danio rerio*) has been widely used in various field of biomedical research to study adult vertebrate regeneration, cancer, metabolic and neurodegenerative diseases [33,34,35]. The adult zebrafish brain is similar to that of other vertebrates in the overall structural components and organization: it displays a similarity to the human brain, notably in defined areas such as the hypothalamus and olfactory bulb. In contrast to mammals, the neurogenic niches occupy widespread regions of the zebrafish brain. One of these niches is the dorsal ventricular zone (VZ) of the telencephalon [36], of which the lateral pallium appears to be homologous to the mammalian hippocampus and amygdala [37]. TBI induction in zebrafish results in a rapid proliferative response leading to neuronal regeneration [38]. This neurogenesis is preceded by an increase in the level of brain-derived neurotrophic factor (BDNF) expression [39,40,41]. Lesion of the dorsolateral telencephalon can completely recover 35 days after injury with Hu-positive newly generated neurons detected at the injury site [38]. Zebrafish have been increasingly used in behavioral neuroscience due to their developed abilities in spatial navigation, Pavlovian conditioning and non-associative learning. Zebrafish allow the understanding of cognitive and adaptive behavior, learning and overall brain function [42].

The aim of this study was to explore the potential of using saffron extract in treating traumatic brain injury outcomes using an adult zebrafish model of open head injury. We report anxiolytic effect of saffron on anxiety-like behaviors, prevention of fear disinhibition, and improvement of memory performance.

## 2. Results

### 2.1. Saffron Attenuates Anxiety-like Behavior in TBI-Induced Zebrafish Model

Anxiety-like behavior was assessed using novel tank and light/dark tests. Novel tank assay is commonly used to evaluate novelty-evoked anxiety in zebrafish. When placed in a novel tank, the zebrafish tends instinctually to dive to the bottom. After minutes of acclimation, it begins to explore the upper regions of the tank [42]. The TBI-induced group spent less time at the top compared to the control group, and this decrease was statistically significant at day 10 (*p* < 0.05; Figure 1A), along with higher latency to enter the top that was statistically significant at day 3 (*p* < 0.05; Figure 1B). Saffron treatment induced a significant increased time spent at the top of the tank (*p* < 0.05. Figure 1A) and a significantly reduced latency to explore the upper portion of the tank (*p* < 0.001; Figure 1B) at the three time points studied compared to untreated TBI.

Anxiety-like behavior was also examined using the light/dark test. This latter is based on the innate preference of adult zebrafish for a black compartment as a unique protective strategy versus a white compartment (scototaxis) [43]. No significant behavioral differences were noted between the three groups (Figure 2) at 3, 10 and 21 days, post-injury (dpi).

### 2.2. Saffron Prevents Alteration of Post-TBI Fear Processes

In order to assess fear in zebrafish, we established a predator avoidance experiment by presenting a video of a sympatric predator, the Indian leaf fish (*Nandus nandus*), to a single zebrafish in a novel environment. The untreated TBI group showed similar behavior to the control, except at D3 where the TBI group spent significantly more time near the predator (81 s) compared to the control group (42 s) (*p* < 0.05; Figure 3). A significant 50% decrease in the time spent near the predator was observed in the untreated TBI group between 10 dpi and 21 dpi compared to 3 dpi (*p* < 0.01 and *p* < 0.001 respectively; Figure 3). No changes were detected in saffron-treated TBI versus control. 

### 2.3. Saffron Ameliorate Learning and Memory Performance in Induced-TBI Zebrafish

We investigated learning and memory alterations 12 days after injury. Experiments were performed in a custom-built behavioral apparatus (Figure 4A). During training sessions (S1; S2; S3), untreated TBI showed overall a poorer performance compared to control whereas saffron-treated TBI exhibited the same behavior as the control (Figure 4B). Taken together, untreated TBI showed a significant decrease in the number of avoidances by ~2-fold compared to control during the training sessions (*p* < 0.05; Figure 4C). No statistical differences between groups were observed for STM (Figure 4C). A decreased number of avoidances of both TBI groups versus control was observed at LTM (Figure 4C). The number of trials in each session that was necessary for zebrafish to achieve the first successful avoidance was noted. During training sessions, the untreated TBI group first avoided on the 4th trial, whereas the control and saffron-treated TBI groups first avoided on the 2nd trial (Figure 4D, upper graph). Statistical analysis showed no significant difference between the groups during training, STM and LTM (Figure 4D, middle and lower graphs respectively).

## 3. Discussion

Saffron (*Crocus sativus* L.) is one of the most expensive spices in the world, used in traditional folk cooking in some areas of the world. Saffron has also been used as a medicinal plant in traditional medicine for hundreds of years [44,45,46]. People commonly used saffron for depression, anxiety, Alzheimer disease, premenstrual syndrome (PMS), cancer, heart disease and other conditions [47,48,49]. Lebanese saffron is characterized by its high quality and its chemical composition containing large amounts of polyphenolics that confers it high anti-oxidant activity and anti-proliferative abilities [50,51]. The aim of this study is to investigate the effect of Lebanese saffron supplementation on behavioral outcomes in an adult zebrafish model of open-head injury. Our results indicate that TBI induces neurological impairment in injured zebrafish, and that saffron supplementation during the first 5 days following TBI is sufficient to alleviate some of the resulting impairments.

Anxiety is a common comorbidity in people with moderate to severe TBI [52]. Prolonged symptoms may even persist for up to several years post-injury and can influence recovery [53]. TBI-induced zebrafish exhibited increased anxiety compared to control across time, spending significantly less time at the top of the novel tank, along with significantly increased latency to enter the upper regions of the tank. Saffron showed a clear anxiolytic effect as early as 3 dpi during which TBI-treated zebrafish showed a reduced latency to enter the top and increased time spent at the top. This result is in line with studies carried out in mice and rats, showing that saffron aqueous extract, and its active compounds such as safranal and crocin, induced an anxiolytic-like effect [54,55,56]. Research suggests that chronic neuroinflammatory responses to injury may play a role in the development of post-traumatic anxiety [57,58,59,60]. Both aqueous and ethanolic saffron extracts were shown to exert anti-inflammatory effects in induced chronic inflammation in mice [61]. A recent study in a repetitive mild non-invasive traumatic brain injury mouse model showed that saffron extract and crocin suppress TBI-induced inflammatory and oxidative responses by decreasing the levels of IFN-γ, TNF-α, MPO, GSH, and MDA [31]. Furthermore, dysregulation in both excitatory and inhibitory neurotransmission has been shown after TBI [62]. Studies have revealed that saffron is implicated in the excitatory/inhibitory balance in the brain, which is crucial for normal functioning [63]. Indeed, saffron inhibits monoamine oxidase (MAOA) enzyme in the synaptic cleft, leading to an increase in serotonin, norepinephrine and dopamine levels which have a pivotal role in the regulation of arousal, mood and anxiety [64]. In addition, saffron modulates the hypothalamic–pituitary–adrenal (HPA) axis activity by reducing plasma corticosterone concentrations [65]. Overall, saffron’s anxiolytic effect could be due to its ability to increase the brain anti-oxidant defenses, and its specific effect on neurotransmitter uptake, receptor modulation and enzyme modulation, primarily MAO and acetylcholinesterase (AChE), that inhibits excitatory/inhibitory imbalance.

Accumulating evidence indicates that neurotrophins via TrK receptors could regulate regenerative processes [41,66]. BDNF mRNA is increased in neurons and in newborn neurons of the injured area after TBI in telencephalon of adult zebrafish, and this increase is maintained up to 15 dpi [39]. Studies imply that saffron anti-depressant and cognitive effects could be mediated via neurotrophin upregulations such as BDNF, neurotrophin-3 (NT-3) and the neurotrophin-inducible neuropeptide VGF [67,68]. The anxiolytic effect of saffron in the novel tank test when comparing treated TBI to untreated TBI seems exacerbated when comparing treated TBI to control. This result may be due to the enhancement of saffron’s effect by upregulation of neurotrophins inherent to TBI.

Unlike in the novel test, neither anxiety-like behavior nor the anxiolytic effect of saffron could be observed in the light and dark test. Interestingly, no other studies on TBI using a zebrafish model measured anxiety by using the light and dark test. This inconsistency observed between the novel tank and light/dark tests in our study underlines the complexity of measuring anxiety-like behavior in animal models. These tests do not measure the same aspect of anxiety-like behaviors [69]. In the novel tank, the motivation for dwelling at the bottom is to escape the surface of the water, and this behavior might reflect not only anxiety but also a combination of locomotor and motivational effects. The light/dark test seems to produce measures that reflect the shyness/boldness of zebrafish instead of the exploration/avoidance behavior.

In our study, we took advantage of an invasive zebrafish model of injury in the dorsal telencephalic pallium. Behavioral and developmental studies demonstrated that the dorsomedial and the dorsolateral part of the zebrafish pallium correspond to the mammalian amygdala and hippocampus, respectively [70,71]. Despite the fact that homology is never assumed to be absolute, the lesion generated in the dorsal telencephalon might disrupt several brain-signaling pathways responsible for normal neurological function and could be an important causative factor in the genesis of fear and anxiety. Fear is an essential, yet complex, reaction that all organisms manifest in response to perceived threat. Under physiological conditions, fear reaction begins in the amygdala, a brain structure critically involved in the processing and modulation of emotions. Subsequently, the amygdala activates areas of the brain involved in preparation for motor functions implicated in the “fight or flight” reaction through the HPA axis [72]. In a mouse model of mild TBI, a dysfunction in the amygdala circuitry leading to a decreased threat response was described [73]. Hadj-Bouziane et al. showed that a damage in the amygdala results in an altered modulation of emotional neural processing demonstrated by a total irresponsiveness to threat [74]. Consistent with this observation, our results revealed that TBI-induced zebrafish exhibited a fear-disinhibition state by spending significantly more time near the predator at 3 dpi. Studies have shown that TBI results in dopaminergic [75] cell loss leading to higher impulsivity and aggressivity [76]. Treatment with saffron extract has been shown to increase brain dopamine concentration [77]. This could explain the possible role of saffron in restoring dopamine dysfunction following injury.

Deficits in long-term memory have been reported in TBI clinical studies [78] and using rodent models [79]. TBI-induced zebrafish presented a decreased performance during the learning phase (training sessions S1 to S3) compared to control and needed more attempts to succeed for the first time in each session. However, this deficit did not have any effect on short-term memory (STM). Saffron administration restores the ability for learning with a performance comparable to control, and has no effect on STM. We hypothesize that the deficits in learning observed following TBI are due to defects in working memory. These deficits can be alleviated with the administration of saffron during the first days following the injury. We hypothesize that this effect is due to saffron anti-oxidative and anti-inflammatory effects that were able to alleviate early secondary insult to the brain. Conversely, saffron fails to restore the deficit observed for long-term memory (LTM) where both untreated and treated TBI-induced zebrafish showed a decreased rate of success compared to control suggesting deficits in memory consolidation. This is probably due to the direct damage caused by TBI through shear injury of neurons and neural tissue. Calcium imaging studies showed that the Dorsomedial part of the telencephalon is activated during aversive reinforcement learning and that activated neurons in the identified area were crucial for long-term memory retrieval [80].

Neurogenesis is a helpful process through which organisms replace their injured neurons with newly generated ones [81]. In contrast to mammals, zebrafish rapidly activate neurogenesis following a TBI. During the first 3 days, Notch-1 signaling is involved in the production of neural precursor cells (NPCs) that later migrate to the lesion then differentiate into mature neurons [38]. A study by Wang et al. [17] showed that inhibition of Notch signaling significantly decreased the protective effects of crocin on neuronal apoptosis, microglial response, and release of pro-inflammatory cytokines. We hypothesize that saffron’s effect on secondary insult post-TBI might be enhanced by this increased Notch-1 signaling inherent to the zebrafish model. The use of this model with highly regenerative capacity could be a valuable tool to study the interplay between saffron and the Notch pathway in the activation of a regenerative response.

We consider some methodological limitations in this study and discuss aspects that can be considered for further investigation. First, we acknowledge that using both male and female zebrafish in each group could hide sexually dimorphic responses that could be noticed [82,83]. Genetics and sex hormones are likely to influence inflammation, edema, oxidative stress, excitotoxicity, and mitochondrial function. Second, behavioral testing described in this study demonstrates a beneficial effect of saffron on TBI recovery and uncovers molecular targets and pathways that are worth being investigated. Possible future experiments to monitor the effect of saffron would be to assess serotonergic function/activity following concussion using calcium imaging. To assess saffron anxiolytic effect through the HPA axis, secretions of cortisol in blood could be monitored throughout recovery. To look at the role of neurotrophin/Trk pathways, mRNA expression levels and protein levels of growth factors such as BDNF and GDNF, their receptors (TrkB, Gfrα), and components of neurotrophin-dependent signaling pathways such as PI3k/Akt, PLCγ/PKC, PLCγ/CAMKII, Ras-Erk1/2 and Rac1-Cdc42 may be assessed with and without saffron treatment. To study the possible interplay between saffron and the Notch pathway, targeting Notch signaling with an antagonist and investigating expression of Notch target genes after saffron treatment may indicate a synergetic effect. Furthermore, immunostainings could be performed to track cell death, especially in dopaminergic neurons, and neural regeneration.

In conclusion, the present work adds evidence to the beneficial effects of saffron as a safe therapeutic option that can be developed to treat TBI. We have shown that saffron extract can reduce mood deregulations, and improve the management of cognitive and learning defects arising after injury. Further studies are needed to understand how saffron can modulate the neurobiological systems that underlie these defects.

## 4. Materials and Methods

### 4.1. Animals and Housing

All experimental protocols were approved by the Lebanese University ethics committee and adhered to the Guide for the Care and Use of Laboratory Animals published by the US National Research Council committee. Thirty-four adult male and female *Danio rerio* zebrafish, aged 6 to 12 months old, were obtained from a specialized commercial supplier UMS AMAGEN CNRS INRA (Gif-sur-Yvette, France). Fish were maintained under standard conditions at 28 °C and raised in a 14 hr/10 hr light/dark cycle. Zebrafish were housed in tanks with constant aeration and water renewal. Animals were fed twice a day with Tetramin Tropical Flakes (Tetra, Blacksburg, VA, USA).

### 4.2. Saffron Extract Preparation

*Crocus sativus* L. (saffron) harvested from the Bekaa valey—Lebanon, were extracted [84]. Two grams of dried stigmas were suspended in a 40 mL mixture of methanol and water (50:50, *v*/*v*) and stirred for 24 hr at 4 °C in a dark room. The solution was filtered and the methanol was evaporated at 40 °C using a rotavapor. The obtained solution was subjected to refrigeration before lyophilization. Mass yield of saffron extract was 8% *w*/*w*.

### 4.3. Traumatic Brain Injury Induction

Adult zebrafish were acclimatized in the room 4–5 days prior to experiment. A total of 34 fish were randomly divided into 3 groups (1) control (CTL-saffron; *n* = 14), (2) untreated TBI (TBI-saffron; *n* = 14), (3) treated TBI (TBI + saffron; *n* = 6). All of the manipulations on all 3 groups were performed after anesthetizing the zebrafish in 16 mg/mL tricaine solution (MS222; Sigma-Aldrich, St. Louis, MO, USA) until unresponsive to tail pinch. Anesthetized fish were secured on a block of tricaine-soaked foam set into a petri dish under a dissecting microscope with the light from the top. Treated zebrafish were injected intraperitoneally for 5 consecutive days with 10 µL of 50 mg/kg saffron diluted in Phosphate-buffered saline (PBS), by inserting a 27-G needle into the midline between the pelvic fins. Control group and untreated TBI group were injected daily with PBS. Following the first intraperitoneal injection, the zebrafish from untreated and treated TBI groups was placed in a dorsal position facing the lens of the dissecting microscope. A 27-gauge needle was inserted into the dorsolateral domain of the right telencephalic hemisphere to produce a lesion of around 2 mm depth (Figure 5). Immediately after, the fish was placed back into freshwater to recover.

### 4.4. Behavioral Assessment

Post-injury behavioral testing was performed on zebrafish. These tests emphasize the behavior-related features including stress, anxiety, fear and memory. All procedures were performed in an isolated room and took place between 9 am and 5 pm. To adapt to the environment, fish were transferred to the experiment room one week prior to the tests. Fish were individually tested in Plexiglas tanks designed for each test. The tank water was changed between test for each fish and maintained at a constant temperature 27 °C, which is optimal for the zebrafish. The zebrafish were fed right before the trials, to ensure that their performance was not hindered by hunger. All trials were recorded using a camera (SONY, Tokyo, Japan) positioned facing the long side of the tank. The. novel tank test, light/dark test and predator test were performed on a total of 34 fish. Behavioral assays were executed on 3 dpi, 10 dpi, and 21 dpi. The avoidance test was performed on a total of 22 fish at day 12 post-lesion.

### 4.5. Novel Tank Test

The apparatus of the test was a trapezoidal tank, 22.5 cm along the bottom, 29.3 cm along the top, 15.5 cm high and 16.2 cm along the diagonal side. Each fish was individually placed in the experimental tank filled with water up to 15 cm and virtually divided into two horizontal zones. The behavioral responses were immediately recorded during a single 9-min testing session. Data collection began after a 1-min habituation period. Following the tank dive, the fish were removed from the experimental tank and placed back into the holding tank. To evaluate exploratory behavior and locomotion of all zebrafish groups, we measured endpoints including time spent at the top (s) and latency to enter the top (s).

### 4.6. Light and Dark Test

The tank used for the light/dark test was a rectangular apparatus (45 cm length × 10 cm width × 15 cm height) and was divided equally into two compartments: one half was covered with black cardboard on all sides of the tank, leaving a small window at the bottom, and the other half was covered with white cardboard on the walls and floor. Each fish was transferred individually into the white compartment of the tank. Following a 3-min acclimation, behaviors were scored over 5 min. Preference for light versus dark side was quantified by assessing the time spent at the light area (s).

### 4.7. Predator Test

Fear and escape in zebrafish were assessed by presenting a video of a sympatric predator, the Indian leaf fish (*Nandus nandus*) to a single zebrafish in a novel environment. Normally, zebrafish restrict swimming to the end of the tank and remain far from the stimulus. The test apparatus consisted of a rectangular tank (35 cm length × 10 cm width × 15 cm height). A video showing an Indian leaf fish was presented on one side of the tank during the total test duration. The tank was divided into two virtual vertical zones: the closest zone to the predator represented 1/3 of the tank where the video was presented, and the other 2/3 of the tank was considered the “safe” zone. Each fish was placed individually at the opposite side of the video in the “safe” zone. Time spent near the predator (s) was measured over a 5-min period.

### 4.8. Active Avoidance Test

In order to assess memory in zebrafish, experiments were performed in a custom-built behavioral setup (see Figure 5A). The apparatus consisted of a rectangular tank (45 cm length × 10 cm width × 15 cm height) with a hurdle in its middle. Four conductive sheet metals were placed opposite, on each side of the four walls, and were attached to a 12 V stimulator that administered a final 6 ± 0.2 V shock when manually activated. Two green LED lights were positioned at both sides of the tank. Zebrafish placed in a beaker underwent a pre-exposure session consisting of 15 trials over 8 min where green light was followed by a brief electric shock. Afterwards, the zebrafish was placed in the tank without any stimulus, for a 5-min acclimation period. The training procedure consisted of three sessions of 12 trials each separated by an inter-trial interval (ITI) of 2 min. A trial consisted of a conditioned stimulus (CS) (green light) given for 10 s in the occupied chamber of the tank. If the fish did not escape the chamber, an electric shock (unconditioned stimulus-US) of 5 s was given and the trial was considered as unsuccessful. The CS-US interval was 10 s. When the fish escaped the chamber before an electric shock was given, the trial was considered as successful and was noted as “avoidance”. Following each session, the zebrafish was left to rest for 4 min. At the end of the training session, each fish was placed back in the holding tank. Short-term memory (STM) was assessed 30 min after the 3rd session of the learning phase. Long-term memory (LTM) was tested 24 h after the short memory test. Both STM and LTM tests consist of a single 7-min session of 12 trials. Endpoints recorded during all phases of the actual test were number of avoidances. Only the first 6 trials over 12 were used in the analysis to minimize fatigue-related bias. The average of the number of avoidances for the 3 first sessions was calculated and indicated as training sessions.

### 4.9. Statistical Analysis

All quantitative data were analyzed using Prism software and were presented as mean ± SEM. Results were compared by a two-way ANOVA followed by Tukey’s test for multiple comparisons used for significance between different groups. Significance was set at *p* < 0.05.

## Figures and Tables

**Figure 1 ijms-23-11600-f001:**
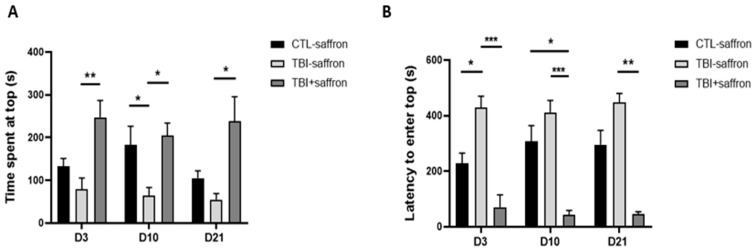
Effect of saffron on induced-TBI zebrafish behavior in novel tank diving test on days 3, 10 and 21 post-lesion. (**A**) Time spent at top of the tank (s) in CTL-saffron (*n* = 14), TBI-saffron (*n* = 14), and TBI + saffron (*n* = 6) at 3-, 10- and 21-days post-injury. (**B**) Latency to enter the top (s) in CTL-saffron (*n* = 14), TBI-saffron (*n* = 14), and TBI + saffron (*n* = 6) at 3-, 10- and 21-days post-injury. Statistical analysis was performed with two-way ANOVA. Statistically significant differences between groups are indicated as * *p* < 0.05, ** *p* < 0.005, *** *p* < 0.001. All data represent the mean ± S.E.M.

**Figure 2 ijms-23-11600-f002:**
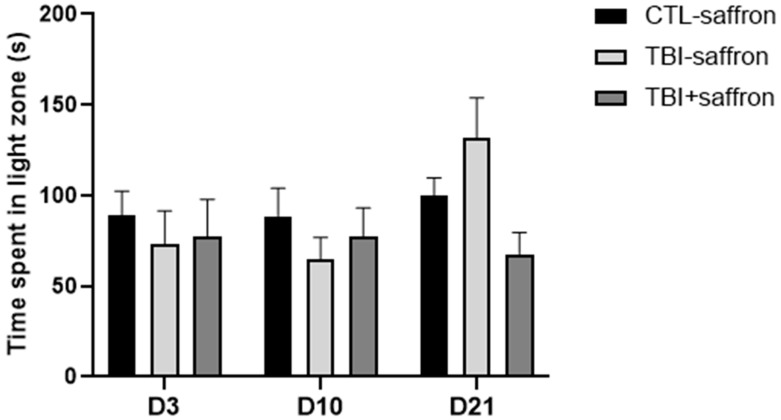
Effect of saffron on induced-TBI zebrafish behavior in the light/dark test on days 3, 10 and 21 post-lesion. Time spent in the light area (s) in CTL-saffron (*n* = 14), TBI-saffron (*n* = 14), and TBI + saffron (*n* = 6) at 3-, 10- and 21-days post-injury. No statistically significant differences were noted using a two-way ANOVA analysis. All data represent the mean ± S.E.M.

**Figure 3 ijms-23-11600-f003:**
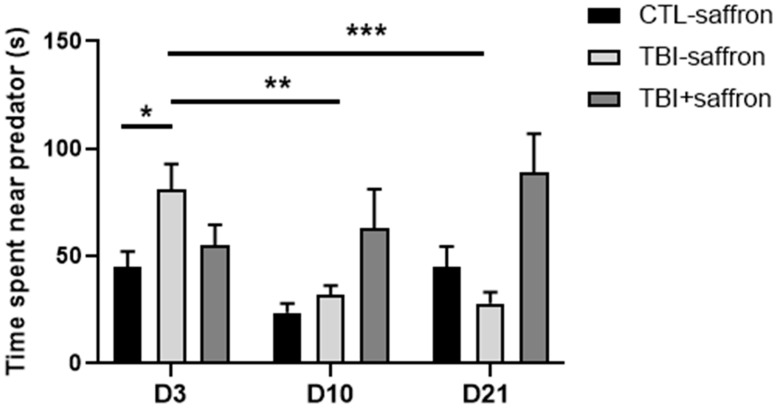
Effect of saffron on induced-TBI zebrafish behavior in the predator test on days 3, 10 and 21 post-lesion. Time spent near the predator comparisons between CTL-saffron (*n* = 14), TBI-saffron (*n* = 14), and TBI + saffron (*n* = 6) at 3-, 10- and 21-days post-injury. Statistical analysis was performed with two-way ANOVA. Statistically significant differences are indicated as * *p* < 0.05, ** *p* < 0.005, *** *p* < 0.0001. All data represent the mean ± S.E.M.

**Figure 4 ijms-23-11600-f004:**
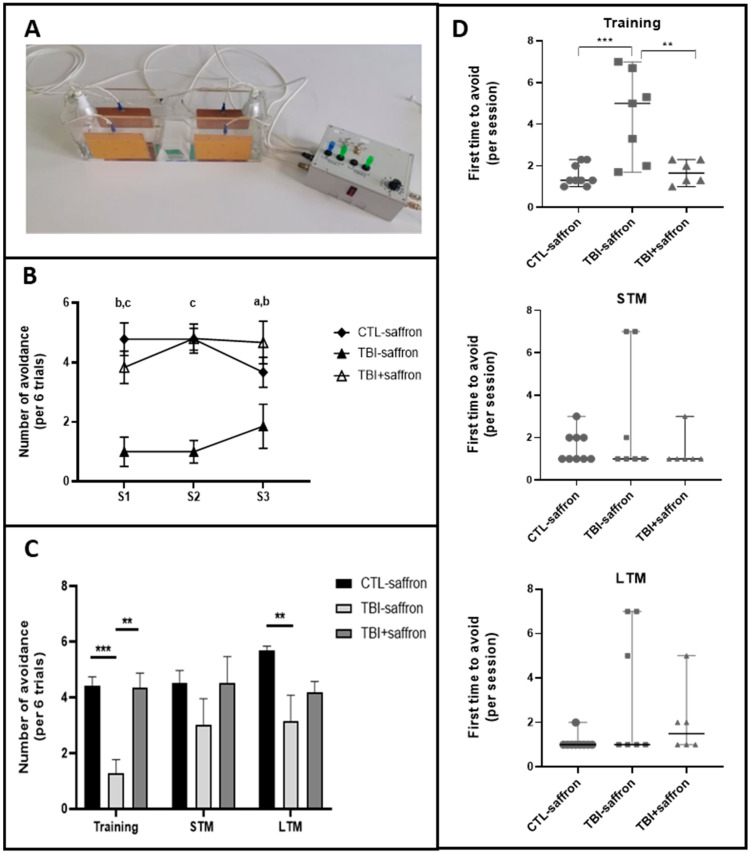
Active avoidance test on zebrafish 12 days post-injury to evaluate the effect of saffron on learning, short-term memory (STM) and long-term memory (LTM). (**A**) Experimental set-up used for active avoidance test. (**B**) Comparison of the number of avoidance (per 6 trials) in CTL-saffron (*n* = 9), TBI-saffron (*n* = 7), and TBI + saffron (*n* = 6) at 12 days post-injury during training session S1, S2 and S3. Statistical analysis was performed with two-way ANOVA. Statistically significant differences are indicated as a (in S1: CTL-saffron vs TBI-saffron; *p* < 0.05); b (in S1: TBI + saffron vs TBI-saffron; *p* < 0.005); c (in S1: CTL-saffron vs TBI-saffron; *p* ≤ 0.0001 and in S2: CTL-saffron and TBI + saffron vs TBI-saffron; *p* ≤ 0.0001). Data represent the mean ± S.E.M. (**C**) Comparison of the number of avoidance (per 6 trials) in CTL-saffron (*n* = 9), TBI-saffron (*n* = 7), and TBI + saffron (*n* = 6) at 12 days post-injury during training sessions (average number during S1, S2 and S3 taken together), STM and LTM. Statistical analysis was performed with two-way ANOVA. Statistically significant differences are indicated as ** *p* < 0.005, *** *p* < 0.0001. Data represent the mean ± S.E.M. (**D**) Comparison of the number of trials (per session) that was necessary for zebrafish to effectuate the first successful avoidance in CTL-saffron (*n* = 9), TBI-saffron (*n* = 7), and TBI + saffron (*n* = 6) at 12 days post-injury during training sessions (average number during S1, S2 and S3 taken together; upper graph), STM (middle graph) and LTM (lower graph). Statistical analysis was performed with one-way ANOVA. Statistically significant differences are indicated as ** *p* < 0.005, *** *p* < 0.0001. Data represent the median ± S.E.M.

**Figure 5 ijms-23-11600-f005:**
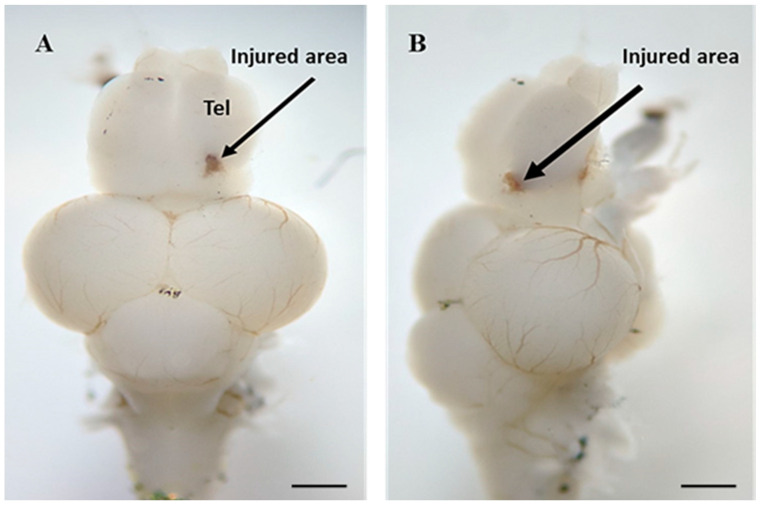
(**A**) Dorsal and (**B**) lateral view of an adult zebrafish brain illustrating the injury site (black arrow) in the right dorsal telencephalon. Tel, Telencephalon. Scale bar: 1 mm.

## Data Availability

The data presented in this study are available on request from the corresponding author.

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
