# Peer review of "Saffron Extract Attenuates Anxiogenic Effect and Improves Cognitive Behavior in an Adult Zebrafish Model of Traumatic Brain Injury"

_ijms, 2022, doi:10.3390/ijms231911600_

Round 1
Reviewer 1 Report
From my chair, it is obvious that this is not a "molecular" study.
In my opinion it is an editorial decision to which journal this fine paper is suitable.
Author Response
Comments and Suggestions for Authors
From my chair, it is obvious that this is not a "molecular" study.
In my opinion it is an editorial decision to which journal this fine paper is suitable.
We thank you for your consideration of our manuscript. We concur that the behavioral part is prominent in the manuscript. The general findings reflect molecular mechanisms, and we tried with careful review of literature to identify pathways and key molecular targets that can be relevant to TBI and treatments. We hope you can consider the revised version.
Reviewer 2 Report
Review of manuscript entitled: “Saffron extract attenuates anxiogenic effect and improves cognitive behavior in an adult zebrafish model of traumatic brain injury” authored by Victoria Chaoul, Maria Awad, Frederic Harb, Fadia Najjar, Aline Hamade, Rita Nabout, Jihane Soueid
First of all I want to thank you for opportunity to review this interesting manuscript.
In the presented article, authors investigated effect of intraperitoneal administration of saffron extract on anxiety-like and cognitive behaviors in zebrafish subjected to traumatic brain injury (TBI). Authors showed that administration of saffron extract provides some beneficial effects against TBI.
Introduction is well-written and provides basic information about undertaken problem. Methods section is written comprehensively, yet some improvements have to be made. Results are presented clearly. Discussion and conclusions are written based on obtained results.
Overall, manuscript is good and written nicely but unfortunately in my opinion it cannot be published in International Journal of Molecular Sciences. I appreciate effort put in behavioral testing and I can imagine amount of work needed to perform all experiments.
Below I provide my remarks and justification of my decision.
Major concerns:
- From IJMS website we can read “(…)provides an advanced forum for molecular studies in biology and chemistry, with a strong emphasis on molecular biology and molecular medicine.”, yet presented manuscript is purely behavioral, I cannot see single molecular analysis here. I suggest changing the journal (for example Biomedicines or Brain Sciences).
- In discussion paragraph authors hypothesize about several things (e.g. Notch signaling pathway, HPA axis etc.) but did not investigate it in their research, this could be a great addition to your work and would make your work suitable to apply for IJMS.
- I do not understand lack of “CTL + Saffron” group. Since you observe that administration of saffron extract in TBI group lowers latency to enter top below the control group it would be beneficial to have “CTL + Saffron” group. Also we can observe that time spent at top may be also hard to interpret since “TBI + Saffron” is much longer than “CTL – Saffron” in D3 and D21.
- Also abovementioned effects are not discussed in discussion paragraph.
- Methods section – authors stated that “Thirty-four adult male and female Danio rerio zebrafish, aged 6 to 12 months old, were obtained from a specialized commercial supplier UMS AMAGEN CNRS INRA (France).”. Such difference in age could have impact on your results, particularly in memory testing.
- Moreover, there is no information provided for sex ratio in each group.
- Authors stated that “All of the induced injuries were performed after anesthetizing the zebrafish in 16 mg/ml tricaine solution (…)” Did control group also receive anesthesia without TBI induction?
- Did authors perform chromatography to asses ratio of four main components mentioned in introduction (crocin, picrocrocin, safranal and crocetin) in obtained saffron extract?
Minor concerns:
- Line 244 – “(…) deficit didn’t have (…)” – please try to avoid abbreviations like this – should be “did not”
Author Response
Comments and Suggestions for Authors
Review of manuscript entitled: “Saffron extract attenuates anxiogenic effect and improves cognitive behavior in an adult zebrafish model of traumatic brain injury” authored by Victoria Chaoul, Maria Awad, Frederic Harb, Fadia Najjar, Aline Hamade, Rita Nabout, Jihane Soueid
First of all I want to thank you for opportunity to review this interesting manuscript.
In the presented article, authors investigated effect of intraperitoneal administration of saffron extract on anxiety-like and cognitive behaviors in zebrafish subjected to traumatic brain injury (TBI). Authors showed that administration of saffron extract provides some beneficial effects against TBI.
Introduction is well-written and provides basic information about undertaken problem. Methods section is written comprehensively, yet some improvements have to be made. Results are presented clearly. Discussion and conclusions are written based on obtained results.
Overall, manuscript is good and written nicely but unfortunately in my opinion it cannot be published in International Journal of Molecular Sciences. I appreciate effort put in behavioral testing and I can imagine amount of work needed to perform all experiments.
We thank you sincerely for your support to our work. Your remarks are very helpful to us in revising our manuscript.Below I provide my remarks and justification of my decision.
Major concerns:
From IJMS website we can read “(…) provides an advanced forum for molecular studies in biology and chemistry, with a strong emphasis on molecular biology and molecular medicine.”, yet presented manuscript is purely behavioral, I cannot see single molecular analysis here. I suggest changing the journal (for example Biomedicines or Brain Sciences).
In discussion paragraph authors hypothesize about several things (e.g. Notch signaling pathway, HPA axis etc.) but did not investigate it in their research, this could be a great addition to your work and would make your work suitable to apply for IJMS.
We agree. We added a paragraph to the discussion about aspects that can be considered for further investigation (lines 279 to 295). We concur that the behavioral part is prominent in the manuscript. The general findings reflect molecular mechanisms, and we tried with careful review of literature to identify pathways and key molecular targets that can be relevant to TBI and treatments.
• I do not understand lack of “CTL + Saffron” group. Since you observe that administration of saffron extract in TBI group lowers latency to enter top below the control group it would be beneficial to have “CTL + Saffron” group. Also we can observe that time spent at top may be also hard to interpret since “TBI + Saffron” is much longer than “CTL – Saffron” in D3 and D21.
Also abovementioned effects are not discussed in discussion paragraph.
We agree that a group control+saffron is missing. The main objective of the study was to investigate the effect of saffron on TBI recovery, and with a limited number of zebrafish that was available, we decided to look at the 3 groups included in the study. Nevertheless, as pointed out by the reviewer, we added a paragraph to discuss the exacerbated anxiolytic effect of saffron in treated-TBI compared to control (lines 218 to 225). After reviewing the literature, we hypothesize that this effect is mediated by an upregulation of neurotrophins observed post-TBI.
• Methods section – authors stated that “Thirty-four adult male and female Danio rerio zebrafish, aged 6 to 12 months old, were obtained from a specialized commercial supplier UMS AMAGEN CNRS INRA (France).”. Such difference in age could have impact on your results, particularly in memory testing. Moreover, there is no information provided for sex ratio in each group.
According to ZFIN website, adult zebrafish is considered from 90 days to 2 years of age. In a study on aging and exercising, Gilbert et al. shows that zebrafish motor skills decrease after 25 months of age. Yu et al. shows that age-related changes in zebrafish cognition manifest at 2 years of age and then further deteriorate.
We didn’t look at the sex ratio in each group, and we agree that this could hide sexually dimorphic responses that could be noticed. We added this remark as limitations in the discussion (lines 280 – 281). Thank you.
REFERENCES:
1- Gilbert MJ, Zerulla TC, Tierney KB. Zebrafish (Danio rerio) as a model for the study of aging and exercise: physical ability and trainability decrease with age. Exp Gerontol. 2014 Feb;50:106-13. doi: 10.1016/j.exger.2013.11.013. Epub 2013 Dec 3. PMID: 24316042.
2- Yu L, Tucci V, Kishi S, Zhdanova IV. Cognitive aging in zebrafish. PLoS One. 2006 Dec 20;1(1):e14. doi: 10.1371/journal.pone.0000014. PMID: 17183640; PMCID: PMC1762370.
• Authors stated that “All of the induced injuries were performed after anesthetizing the zebrafish in 16 mg/ml tricaine solution (…)” Did control group also receive anesthesia without TBI induction?
All zebrafish from the 3 groups received anesthesia. TBI groups received Control and untreated-TBI groups received PBS by intraperitoneal injections during the first 5 days, whereas treated-TBI group received saffron. Following the first injection, only TBI groups received needle injury in the telencephalon. We corrected the paragraph in method section (lines 324 – 326).
• Did authors perform chromatography to asses ratio of four main components mentioned in introduction (crocin, picrocrocin, safranal and crocetin) in obtained saffron extract?
We couldn’t perform chromatography. But this must definitely be investigated, especially with molecular studies to investigate how saffron can modulate the neurobiological systems that underlie TBI defects. We thank the reviewer for this remark
Minor concerns:
Line 244 – “(…) deficit didn’t have (…)” – please try to avoid abbreviations like this – should be “did not”
We corrected all abbreviations in the text.
Reviewer 3 Report
The article presented from Victoria Chaoul et al., entitled "Saffron extract attenuates anxiogenic effect and improves cognitive behavior in an adult zebrafish model of traumatic brain injury" is interesting. However there are several critical points to address:
- The authors should improve the introduction and discussion including relevant references in the field. In detail the authors does not mention more recent studies on TBI in adult zebrafish. At the same time they should also insert which are the genetic and molecular pathways involved after TBI (on telencephalon). For reference see:
- Cacialli, Pietro; Lucini, Carla. Adult neurogenesis and regeneration in zebrafish brain: are the neurotrophins involved in?. Neural Regeneration Research: December 2019 - Volume 14 - Issue 12 - p 2067-2068 doi: 10.4103/1673-5374.262574
- Cacialli, P. Neurotrophins Time Point Intervention after Traumatic Brain Injury: From Zebrafish to Human. Int. J. Mol. Sci. 2021, 22, 1585. https://doi.org/10.3390/ijms22041585
- Anand, S. K., Sahu, M. R., & Mondal, A. C. (2021). Induction of oxidative stress and apoptosis in the injured brain: potential relevance to brain regeneration in zebrafish. Molecular biology reports, 48(6), 5099–5108. https://doi.org/10.1007/s11033-021-06506-7
Concerning the results I would suggest the authors to confirm their TBI model:
- They should show by using histology and immunohistochemistry the lesion on telencephalic region.
In the abstract, introduction, results and discussion the authors write "These results suggest that saffron could be a novel therapeutic enhancer for neural repair and regeneration of networks post-TBI".
But it is not, the conclusions are not supported by their results. To validate the role of this drug during the neuro-regeneration (if it has) the authors should perform
qPCR, in situ hybridization and or immunohistochemistry to evaluate the levels of several neural stem cell markers (PCNA, GFAP, Aromatase, Nestin etc)
This study is interesting but only supported by Behaviour tests.
-
Author Response
Comments and Suggestions for Authors
The article presented from Victoria Chaoul et al., entitled "Saffron extract attenuates anxiogenic effect and improves cognitive behavior in an adult zebrafish model of traumatic brain injury" is interesting. However, there are several critical points to address:
We thank you for your consideration of our manuscript. Based on your suggestions, we have made modifications to improve the manuscript.
The authors should improve the introduction and discussion including relevant references in the field. In detail the authors does not mention more recent studies on TBI in adult zebrafish. At the same time they should also insert which are the genetic and molecular pathways involved after TBI (on telencephalon). For reference see:
Cacialli, Pietro; Lucini, Carla. Adult neurogenesis and regeneration in zebrafish brain: are the neurotrophins involved in Neural Regeneration Research: December 2019 - Volume 14 - Issue 12 - p 2067-2068 doi: 10.4103/1673-5374.262574 - Cacialli, P. Neurotrophins Time Point Intervention after Traumatic Brain Injury: From Zebrafish to Human. Int. J. Mol. Sci. 2021, 22, 1585. https://doi.org/10.3390/ijms22041585 - Anand, S. K., Sahu, M. R., & Mondal, A. C. (2021). Induction of oxidative stress and apoptosis in the injured brain: potential relevance to brain regeneration in zebrafish. Molecular biology reports, 48(6), 5099–5108. https://doi.org/10.1007/s11033-021-06506-7
According to reviewer requests, we have added references to the introduction and discussion, and have made extensive modifications on both sections. In the introduction, we added information about neurogenesis occurring after TBI and references. In the discussion, we added a paragraph to discuss the exacerbated anxiolytic effect of saffron in treated-TBI compared to control (lines 218 to 225).
Concerning the results, I would suggest the authors to confirm their TBI model:
- They should show by using histology and immunohistochemistry the lesion on telencephalic region.
We unfortunately do not possess injured brains that were dissected in the days following injury. Remaining brains were collected at the end of the behavioral studies after 21 days. The lesion is most certainly already recovered partially if not completely. We only have a visual of the lesion in figure 5.
In the abstract, introduction, results and discussion the authors write "These results suggest that saffron could be a novel therapeutic enhancer for neural repair and regeneration of networks post-TBI".
But it is not, the conclusions are not supported by their results. To validate the role of this drug during the neuro-regeneration (if it has) the authors should perform qPCR, in situ hybridization and or immunohistochemistry to evaluate the levels of several neural stem cell markers (PCNA, GFAP, Aromatase, Nestin etc).
This study is interesting but only supported by Behaviour tests.
We agree. The general findings reflect molecular mechanisms, and we tried with careful review of literature to identify pathways and key molecular targets that can be relevant to TBI and treatments. We added a paragraph in the discussion section about limitations of the study and aspects that can be considered for further investigation (lines 279 to 295). We hope you can consider the revised version.
Round 2
Reviewer 1 Report
Your response is appropriate
(Your response) We thank you for your consideration of our manuscript. We concur that the behavioral part is prominent in the manuscript. The general findings reflect molecular mechanisms, and we tried with careful review of literature to identify pathways and key molecular targets that can be relevant to TBI and treatments. We hope you can consider the revised version.
Reviewer 2 Report
Authors responded to all my concerns, thank you for clarifying. In my opinion, from scientific point of view everything is fine and the manuscript is well written. I wish you good luck with further work!
The decision whether it suits the IJMS aims and scope belongs to the editor.
Reviewer 3 Report
The authors satisfied all my concerns. I propose the acceptance in the present form.